# Impedimetric DNA Sensors for Epirubicin Detection Based on Polythionine Films Electropolymerized from Deep Eutectic Solvent

**DOI:** 10.3390/s23198242

**Published:** 2023-10-04

**Authors:** Anastasia Goida, Alexey Rogov, Yurii Kuzin, Anna Porfireva, Gennady Evtugyn

**Affiliations:** 1A.M. Butlerov’ Chemistry Institute, Kazan Federal University, 18 Kremlevskaya Street, Kazan 420008, Russia; a.goida@mail.ru (A.G.); yzinkyra@mail.ru (Y.K.); porfireva-a@inbox.ru (A.P.); 2Interdisciplinary Center of Analytical Microscopy, Kazan Federal University, 18 Kremlevskaya Street, Kazan 420008, Russia; alexeyrogov111@gmail.com; 3Analytical Chemistry Department, Chemical Technology Institute, Ural Federal University, 19 Mira Street, Ekaterinburg 620002, Russia

**Keywords:** polythionine, electropolymerization, deep eutectic solvents, electrochemical DNA sensor, electrochemical impedance spectroscopy, epirubicin determination, screen-printed electrode

## Abstract

An electrochemically active polymer, polythionine (PTN), was synthesized in natural deep eutectic solvent (NADES) via multiple potential scans and characterized using cyclic voltammetry and electrochemical impedance spectroscopy (EIS). NADES consists of citric acid monohydrate, glucose, and water mixed in the molar ratio of 1:1:6. Electrodeposited PTN film was then applied for the electrostatic accumulation of DNA from salmon sperm and used for the sensitive detection of the anticancer drug epirubicin. Its reaction with DNA resulted in regular changes in the EIS parameters that made it possible to determine 1.0–100 µM of epirubicin with the limit of detection (LOD) of 0.3 µM. The DNA sensor developed was successfully applied for the detection of epirubicin in spiked samples of artificial and natural urine and saliva, with recovery ranging from 90 to 109%. The protocol of the DNA sensor assembling utilized only one drop of reactants and was performed with a minimal number of steps. Together with a simple measurement protocol requiring 100 µL of the sample, this offers good opportunities for the further use of the DNA sensor in monitoring the drug level in biological samples, which is necessary in oncology treatment and for the pharmacokinetics studies of new antitumor drugs.

## 1. Introduction

Deep eutectic solvents (DES) were first described by Abbott in the early 2000s [1,2]. In the following twenty years, the term DES has changed and is now defined as a mixture of Lewis and Brønsted acids and bases, which significantly reduces the freezing point compared with those of the components. Various aspects of the DES definition are still in progress [3]. The DES classification by the nature of the donor and acceptor was broadened to five classes [4]. Choline derivatives, organic acids, sugars, and alcohols are natural components that offer a wide spectrum of new media suitable for performing electrochemical measurements, including electropolymerization, with no need for organic solvents [5]. The latter advantage is related to better solubility of organic species used as precursors of electropolymerization achieved in DES against that in water. NADES are natural deep eutectic solvents consisting of cell metabolites [5,6]. All these components are biocompatible and take place between the aqueous and lipid phases in living cells [6]. NADES are low-cost and biodegradable solvents; their cytotoxicity depends on components used and can be negligible [7].

Several methods of NADES synthesis are described, i.e., controlled heating and stirring, freeze drying, evaporation, sonication, and microwave-assisted synthesis [8]. As was shown, FTIR spectra, as well as thermal decomposition profiles, showed similarity of the NADES obtained by various methods. No statistically significant differences in the density and viscosity of the products were observed. Although ultrasound- and microwave-assisted NADES syntheses were faster and more efficient, stirring and heating offered easier use, simplicity, and the ability to produce larger volumes of the solvents with reproducible physicochemical characteristics.

Anthracycline drugs, e.g., doxorubicin, daunorubicin, idarubicin, valrubicin, and epirubicin, are widely used in the therapy of various cancer types [9,10]. Meanwhile, their action is often accompanied by dangerous side effects such as high cardiotoxicity. Additionally, while intercalating the DNA, they participate in the inhibition of topoisomerase II activity, generate reactive oxygen species in the presence of molecular oxygen, and can chelate transition metal ions [11]. Epirubicin (Figure 1a) is a semisynthetic derivative of doxorubicin that differs in hydroxyl group configuration at the 4′ position of the aromatic core.

It shows less basic and more lipophilic properties against doxorubicin. Epirubicin hampers the synthesis and function of DNA in cancer cells and is most active within the S phase of the cell cycle [12]. This drug was approved by the FDA in 1999 and is mainly used for brain tumors and auxiliary node–positive breast cancer treatment [9,13,14]. A comparison of equimolar doses of epirubicin and doxorubicin showed less myelosuppression and cardiac toxicity and fewer events of nausea, vomiting, alopecia, and mucositis for epirubicin [15]. For this reason, epirubicin can be prescribed at higher cumulative doses before causing cardiotoxic effects [10,12]. Nevertheless, thorough medication dose control during the chemotherapy period is necessary. Biological liquids analysis can assist in personalized cancer treatment and protect medical staff and pharmacy workers from unintended drug consumption [16]. 

Currently, epirubicin is determined by various physicochemical methods, e.g., spectrofluorometry [17], UV-vis spectroscopy [18], high-performance liquid chromatography (HPLC) with UV [19], fluorometric [20], photodiode array [21], and mass-spectroscopy [22] detectors and electrochemical sensors [23,24,25,26,27,28]. Being sensitive, selective, and universal, chromatography-based approaches require sophisticated equipment and high qualifications of the labor staff and are mostly intended for use in a well-equipped chemical laboratory. Electrochemistry offers an alternative to such equipment, mostly intended for a fast and reliable determination of drug residues in the format of point-of-care testing (POCT) devices [29,30]. Meanwhile, electrochemical sensors detect anthracycline medications, including epirubicin, at rather high working potential. This complicates analysis in biological fluids containing oxidizable organic species. 

The application of DNA as a biological target for antitumor drugs in the assembly of electrochemical biosensors improves their performance and makes it possible to characterize anthracyclines from the point of view of their interaction with DNA [31]. Various types of DNA are commercially available and can be utilized in DNA sensors. They differ in average molar mass and chain length, as well as in the content of single- and double-stranded DNA molecules. In the surface layer, DNA molecules can specifically interact with medical preparations via electrostatic attraction, donor–acceptor interactions, H-bonding, and intercalation mechanisms [32]. Intercalation leads to structural, conformational, charge, and volume changes in the DNA bioreceptor that can be assessed by changes in the charge distribution or permeability of the DNA layer for small redox probes [30]. 

Electropolymerization is one of the promising approaches to assembling DNA sensors for drug determination. Redox-active polymers both accumulate DNA in the layer and transduce the signal of its specific interaction. Polyaniline [33], polypyrrole [34], and polymerized phenothiazine derivatives (Azure A, Azure B, and Methylene blue) [35,36,37,38] were obtained for this purpose. However, the low solubility of the monomers, as well as their high self-aggregation, complicate the deposition of the polymer and influence the metrological characteristics of the biosensors. The use of organic solvents partially compensates for the low monomer concentration available in aqueous solutions but negatively affects the DNA-specific interactions. 

Thionine (Figure 1b) is not as active in electropolymerization as its *N*-methylated analogs, especially Methylene blue, but its polymeric form shows high reversibility of the electron transfer and is applied for the mediated oxidation of small molecules, e.g., dopamine [39] and hydrogen peroxide [40]. Meanwhile, it was never used alone in the DNA sensor intended for the specific drug determination.

In this work, we describe the determination of epirubicin with a DNA sensor based on the polythionine (PTN) electropolymerized in NADES. To the best of our knowledge, this is the first example of the use of DES for the design of electrochemical DNA sensors for drug determination. The use of DES combined with the screen-printed electrode platform made it possible to reduce the quantities of chemicals required for the preparation of the sensing layer and minimize the volume of the samples for antitumor drug determination. The latter is important considering the future application of such DNA sensors in monitoring the drug administration of cancer patients. DES allowed increasing the working concentration of the dye monomer, which is low soluble in water, on the step of electropolymerization. Additionally, the aggregation of the monomers in the reaction media was suppressed. All of these factors promoted electropolymerization and made clearer the interpretation of the voltammetric curves obtained both on the stage of deposition of the coating and its characterization using cyclic voltammetry. Additionally, NADES allows full exclusion of organic solvents in the framework of the green chemistry paradigm. 

Ultrasound-assisted synthesis of NADES was chosen as the preferential method because of its simplicity, fast preparation of the polymer film, and polymerization efficiency. Double-stranded DNA from salmon sperm was physically immobilized on the PTN coating and used as bioreceptor-binding epirubicin. Target interaction was monitored by electrochemical impedance spectroscopy (EIS) following changes in the charge transfer resistance. Simple and express protocol of signal measurement can find a widespread application in clinical and pharmacokinetics studies.

## 2. Materials and Methods

### 2.1. Reagents

Thionine acetate (dye content > 85%), DNA from salmon sperm (Na salt, low mol. weight, ~2000 bp, <5% protein, A_260/280_ = 1.4), DNA from calf thymus (Na salt, A_260_/_280_ ≥ 1.8, mol. weight 15.4–17.4 MDa), bovine serum albumin (BSA, lyophilized powder, ≥96%), and HEPES (N-(2-Hydroxyethyl)piperazine-N′-(2-ethanesulfonic acid)) were purchased from Sigma-Aldrich (Darmstadt, Germany). Glucose, citric acid monohydrate, and epirubicin hydrochloride (purity 98.2%) were purchased from Alfa Aesar (Ward Hill, MA, USA). Other reagents were of analytical grade. Deionized Millipore Q^®^ water (Simplicity^®^ water purification system, Merck-Millipore, Mosheim, France) was used for the preparation of working solutions. Britton-Robinson buffer containing 0.04 M H_3_BO_4_, 0.04 M H_3_BO_3_, 0.04 M CH_3_COOH, 0.05 M Na_2_SO_4_, and pH = 2.0–9.0 was used for the pH dependencies study. Other electrochemical investigations were performed in 0.1 M HEPES containing 0.03 M NaCl, pH 7.0. Electrochemical quartz crystal microbalance (EQCM) measurements were performed in 0.1 M HEPES buffer containing 0.03 M NaNO_3_, pH 7.0. 

### 2.2. Apparatus

Voltammetric and EIS measurements were performed at ambient temperature using potentiostat-galvanostat AUTOLAB PGSTAT 302N with FRA2 module (Metrohm Autolab b.v., Utrecht, the Netherlands). Screen-printed carbon electrodes (SPEs) were produced on the DEC 248 printer (DEK, London, UK) on Lomond PE DS Laser Film (thickness 125 μm, Lomond Trading Ltd., Douglas, Isle of Man). Conducting tracks were printed with the PSP-2 silver paste (Delta-Paste, Moscow, Russia) and carbon tracks with carbon/graphite paste C2030519P4 (Gwent group, Pontypool, UK). The insulating layer was made of dielectric paste D2140114D5 (Gwent group). Each layer was hardened at 80 °C. The electrode group had dimensions of 11 × 27 mm with a geometric working area of the working electrode equal to 3.8 mm^2^. The SPEs were connected with AUTOLAB PGSTAT potentiostat using a boxed connector of the bipotentiostat-galvanostat μStat 400 Metrohm DropSens (DropSens, S.L., Asturias Llanera, Spain). Cyclic voltammetry (CV) and EIS were used for electrochemical characterization of the PTN layer.

The EIS frequency varied from 100 kHz to 0.04 Hz, the amplitude of the applied sine potential was equal to 5 mV, and impedance equilibrium potential was calculated as a half-sum of the peak potentials recorded in a 0.01 M [Fe(CN)_6_]^3−/4−^ pair as a redox probe. The impedance parameters were calculated from the Nyquist diagram with the R(R_1_C_1_)(R_2_C_2_) equivalent circuit using NOVA software (Metrohm Autolab b.v., Utrecht, The Netherlands).

EQCM measurements were performed with the EQCM module of the CHI 440B electrochemical analyzer (CH Instruments, Inc., Austin, TX, USA) equipped with the EQCM AT-cut piezo crystal (fundamental frequency 8 MHz, diameter of working thin gold electrode: 0.205 cm^2^).

Scanning electron microscopy (SEM) images were obtained using the Metrohm DropSens DRP-110 screen-printed carbon electrodes as supports (DropSens, S.L., Asturias Llanera, Spain). The Merlin™ high-resolution field emission scanning electron microscope (Carl Zeiss AG, Oberkochen, Germany) was used with the ZeissSmartSEM software (V05.06).

The HPLC determination of epirubicin in spiked saliva was performed using the Agilent 12000 HPLC system (Agilent Technologies, Inc., Santa Clara, CA, USA) with a diode array detector (45/55 phosphate buffer/methanol, 1 mL/min, isocratic elution, 40 °C) [21].

Statistical data treatment was performed using OriginPro 8.1 software (OriginLab Corp., Northampton, MA, USA).

### 2.3. Polythionine Electropolymerization and DNA Sensor Assembling

NADES was prepared by mixing 0.21 g of citric acid monohydrate and 0.18 g of glucose with 90 µL of deionized Millipore Q^®^ water. NADES containing thionine was prepared from 0.21 g of citric acid monohydrate, 0.18 g of glucose, and a certain amount (1.7, 3.4, 6.8, or 10.2 mg) of thionine acetate mixed with 90 µL of deionized Millipore Q^®^ water. The molar ratio of the mixtures was equal to 1:1:6 for citric acid, glucose, and water, respectively. The thionine concentration in NADES corresponded to the addition of 10.2 mg of the dye, which was equal to 0.1 M. NADES was homogenized using a vortex for 1 min, followed by sonication for 30 min. Then, 100 µL of the NADES were drop casted on the SPE surface to entirely cover the three-electrode system, and the background voltammogram was recorded in the range from −0.6 to 0.8 V with a scan rate of 100 mV/s. Then, the NADES drop was washed off with distilled water, and 100 µL of the NADES containing thionine were drop casted on the SPE surface (Appendix A). The potential of the working electrode was 20 times cycled between −1.2 and 1.2 V with a scan rate of 100 mV/s. The appropriate coating is denoted below as PTN_NADES_.

Then, the electrode was stabilized in open circuit mode. For this purpose, the electrode with PTN_NADES_ layer was disconnected from the potentiostat; 100 µL of working buffer was drop casted onto the three-electrode system surface for 30 min. After that, a voltammogram was recorded between −0.6 and 0.6 V with a scan rate of 100 mV/s. DNA from salmon sperm was immobilized over the PTN_NADES_ layer by physical adsorption. An amount of 2 µL of the 1 mg/mL DNA solution was placed onto the working surface and capped with plastic tubing for 20 min to prevent drying. 

As a reference, SPE covered with the PTN obtained from the buffer solution and stabilized as described above was considered. The optimal thionine electropolymerization conditions from aqueous media were determined in [38]. First, 100 µL of 0.1 M HEPES containing 0.03 M NaCl, pH 7.0, were placed on the SPE surface, and the background voltammogram was recorded as described above. Then, the electrode was washed with deionized water, and 100 µL of a fresh buffer containing 0.1 mM thionine was drop casted on the SPE surface. The potential of the electrode was 20 times cycled between −0.5 and 1.1 V with a scan rate of 100 mV/s. The resulting layer is denoted as PTN_aq_. 

### 2.4. Epirubicin Measurements, Real Sample Analysis

An amount of 2 µL of the epirubicin solution (100 nM–100 µM) in 0.1 M HEPES containing 0.03 M NaCl, pH 7.0, was added to the working electrode of the PTN_NADES_/DNA sensor. The electrode was then covered with plastic tubing for 10 min to prevent drying, washed with deionized water, and dried on air at ambient temperature. After that, 100 µL of the same HEPES containing 0.01 M equimolar mixture of [Fe(CN)_6_]^3−/4−^ as a redox probe were added to establish hydrolytic contact of all three electrodes, and EIS measurements were performed.

Ringer-Locke’s solution was taken as an artificial blood plasma sample (9 g/L NaCl, 0.42 g/L KCl, 0.5 g/L NaH_2_PO_4_·2H_2_O, 0.32 g/L CaCl_2_·2H_2_O, 0.1 g/L NaHCO_3_, 0.3 g/L MgSO_4_, 1.5 g/L D-glucose), pH 7.0. A moderate level of albumin typical for adults’ blood serum (41.4 mg/mL bovine serum albumin (BSA)) was chosen for interferences assessment. Artificial urine samples contained 20 mM KCl, 49 mM NaCl, 15 mM KH_2_PO_4_, 10 mM CaCl_2_, 18 mM NH_4_Cl, and 18 mM urea. Human urine and saliva samples were collected from conditionally healthy volunteers. The pH value of artificial and human urine was corrected to 7.0. Urine samples were filtrated through the filtration paper if sediment appeared during the pH correction. Saliva samples were used without pH correction.

## 3. Results and Discussion

### 3.1. Thionine Electropolymerization on Carbon SCE from Aqueous Solution

Electropolymerization of thionine in 0.1 M HEPES containing 0.03 M NaCl, pH 7.0 resulted in characteristic changes in voltammograms (Figure 2). At the first scan, a pair of the monomer redox peaks in the area near −0.3 V and an irreversible oxidation wave at about 0.8 V appeared; the latter one was referred to as the cation radical formation. The relatively low potential of the formation of thionine cation radicals coincides with the behavior of phenothiazines with a primary amino group at the aromatic core [41]. An appropriate wave was reduced with the increasing number of potential scans. Starting from the second cycle, another signal appeared and grew at more positive potentials, reaching 0.85 and −0.155 V in the 20th cycle. Cathodic peaks on voltammograms were recorded at the thionine polymerization and were resolved better than corresponding anodic peaks.

After electropolymerization, a fresh 100 µL drop of 0.1 M HEPES with no thionine was placed on the polymer surface, and the layer was stabilized as described in Section 2.3. After such a treatment, an anodic branch of the voltammogram contained one broadened peak attributed to both polymeric (PTN_aq_) and residual monomeric forms of thionine and the cathodic branch with two overlapped peaks related to both forms of the dye.

In the polymerization, anodic and cathodic currents increased with the number of scans (Figure 3a). After the stabilization step, the redox peaks of the PTN_aq_ layer slightly changed in the opposite direction (see voltammograms for ten consecutive scans in Figure 3b). The cathodic current of the monomeric thionine decreased because of leaching entrapped dye molecules from the layer. Currents of the polymeric form were stabilized more quickly. Changes in voltammograms were more pronounced in the case of the changing buffer drop on the electrode between the scans.

Electropolymerization of thionine in the above conditions was confirmed by EQCM measurement using an Au QCM chip instead of the carbon SPE. Mass changes were recorded simultaneously with the voltammograms, as shown in Figure 4a. Apart from the thionine signals, the cathodic peak at +0.465 V related to the Au oxide reduction can be seen on the voltammograms. Polymerization product accumulation led to an increase in the mass of the surface layer and, respectively, to a decrease in the oscillation frequency measured. The resonant frequency measured at the end of each cycle linearly depended on the cycle number (Figure 4b). A subsequent increase in the number of potential cycles was considered inexpedient because of the elongation of the modification step and feasible overloading of modifier deposited onto the quartz chip, leading to the oscillation freezing.

### 3.2. Thionine Electropolymerization on Carbon SCE from NADES 

The DES possesses a high viscosity and slower diffusion of dissolved species against water. For this reason, the thionine concentration used for electropolymerization in NADES was chosen to be higher (0.1 M) than that used in HEPES. Several parameters (thionine concentration, potential scan rate, and the number of cycles) were varied to establish the optimal experimental conditions (Appendix A). A typical voltammogram recorded in NADES consisted of citric acid, glucose, and water (molar ratio 1:1:6), and 0.1 M thionine is presented in Figure 5. In comparison with that recorded in HEPES (see Figure 2), no irreversible peak responsible for cation-radical formation was found. 

The peak potential difference of the thionine redox conversion (−0.76 and 0.66 V) was higher than that in aqueous media. This could be due to deterioration of the electron exchange in highly viscous solvents. 

After the stabilization of the PTN_NADES_ layer (Section 2.3), two pairs of redox peaks were present on the voltammograms. They were attributed to the monomeric and polymeric forms of thionine. The appropriate currents regularly increased with the thionine concentration changed from 0.017 to 0.1 M. The variation of the currents of the PTN_NADES_ layer was noticeable after 15 cycles of electropolymerization. Meanwhile, the current shift during the electropolymerization delayed after 20 cycles. The potential scan rate varied in range from 50 to 100 mV/s and did not affect the peak currents.

The following increase in the scan rate led to the decay of the signals (Figure 6a–c). Considering the runs of the electropolymerization curves, voltammograms of resulted polymer coatings, and time required for modification time, the following working conditions were specified for the following experiments: 0.1 M thionine, 20 cycles of electropolymerization, scan rate 100 mV/s. Changes in the redox peaks during ten consecutive potential scans for the PTN_NADES_ layer corresponding to these conditions are presented in Figure 6d.

Unfortunately, the high viscosity of NADES suppressed the oscillation of the QCM chip so that the polymerization from this solvent could not be recorded using the EQCM measurements, as in the case of PTN_aq_ film (see Figure 4).

### 3.3. Comparison of the PTN_NADES_/PTNaq Electrochemical Properties

In spite of the difference in the peak currents of the PTN_NADES_ and PTN_aq_ coatings, their pH dependences were compared, and electrochemical characteristics were calculated. Several regimes of the coating stabilization were evaluated, i.e., ten consecutive voltammetric measurements in 100 µL volume of working buffer, ten consecutive voltammetric measurements with fresh 100 µL HEPES volume in each measurement, and open circuit washing of the electrode in 100 µL of the working buffer followed by recording of single voltammogram. Furthermore, coatings with no stabilization of the surface layer were tested in the pH variation experiments. Appropriate dependencies of the peak currents on the pH changed in a similar manner for all mentioned protocols of film stabilization. Eventually, a non-current (open circuit) regime was chosen for film stabilization. 

Both the oxidation and reduction peak currents of the PTN_aq_ decreased monotonously in the studied pH range from 2.0 to 9.0 (Appendix A). The peak current related to the monomeric thionine oxidation depended on the pH both for PTN_NADES_ and PTN_aq_ in the same manner, but the signal stabilization was faster in PTN_NADES_. The PTN_NADES_ polymer oxidation peak current was quite stable in the pH 3.0–7.0 and slightly deviated in more acidic and basic media. The reduction peak current of the PTN_NADES_ attributed to the monomer had a maximum at the pH of about 4.5. A similar dependency for the PTN_NADES_ polymeric form had two maxima at pH 5.0 and 8.0 (Appendix A). The half-sum of the peak potentials for monomeric and polymeric forms of polythionine was used as an estimate of equilibrium potential *E_m_*. The slopes of the linear range of the *E_m_*-pH dependency for the PTN_NADES_ and PTN_aq_ coatings are presented in Table 1. 

The slopes of the pH dependency obtained for the PTN_NADES_ film were close to the theoretical Nernstian value of 59 mV/pH, corresponding to the transfer of an equal number of electrons and hydrogen ions. The PTN_aq_ film exerted a super-Nernstian slope of 79 mV/pH due to possible non-equilibrium conditions of redox exchange in the polymer.

Cyclic voltammograms recorded at different scan rates are presented in Figure 7 for PTN_NADES_ and PTN_aq_ coatings. 

The bi-logarithmic dependency of the peak current (*I_p_*) on the scan rate (ν) allows for establishing the limiting step of the redox conversion (Figs. S7, S8). Electrochemical parameters for the PTN_NADES_ and PTN_aq_ coatings are presented in Table 2. The slope of the bi-logarithmic dependence of the peak current (*I_p_*) on the scan rate (ν) for PTN_NADES_ monomer and polymer oxidation signal and PTN_aq_ redox conversion indicated mixed adsorption-diffusion control (0.78–0.84). The slope of PTN_NADES_ monomer and polymer reduction signal dependence demonstrated adsorption control of the limiting step (1.12 and 0.91, respectively).

The electron transfer coefficient was calculated from the Laviron Equation (1) [42].
(1)$$\frac{dE_{pa}}{d\log(v)}=\frac{2.303RT}{\left(1-\alpha \right)nF};\;\frac{dE_{pc}}{d\log(v)}=-\frac{2.303RT}{\alpha nF}$$

Here, *E_pa_* and *E_pc_* are anodic and cathodic peak potentials, respectively, *R* is a universal gas constant, *T* is absolute temperature, K, *F* is the Faraday constant, and *n* is the number of electrons transferred. Only in the case of the PTN_aq_ oxidation was it in an admissible range from 0.5 to 1.0 (0.83), indicating transfer to a more stable oxidized state. In the Laviron equation, the single elementary step of electrode reaction corresponding to the release and uptake of one electron is taken into account. Actually, the majority of the redox reactions are multi-electron and multi-step processes [43]. Therefore, the electron transfer coefficient exceeding 1.0 reflects the complicated character of the reactions within the PTN layer. 

### 3.4. SEM Characterization of the PTN_NADES_ and PTN_aq_ layers

As was established by SEM, the structure of carbon nanomaterial used for the SPE manufacture was well defined in the absence of a modifying layer (Figure 8a). Electropolymerization led to coverage of the electrode surface with the polymer. Its morphology strictly depended on the electropolymerization media. In the presence of NADES, a folded and quite bulky layer was synthesized (Figure 8b). Contrary to that, electropolymerization from aqueous media led to microgranular structure formation and electrode hollows filling with polymeric material (Figure 8c). The difference might result from the higher viscosity of the monomer dissolved in NADES and irregular access of the thionine molecules to the growing polymer sediment.

### 3.5. DNA Implementation Effects

Double-stranded DNAs from salmon sperm or from calf thymus were physically adsorbed on the PTN_NADES_ layer. An amount of 2 µL drop of appropriate DNA solution with a concentration of 1 mg/mL was either dried on air at ambient temperature or incubated for 20 min under the cap, preventing its drying. The resulting changes in the PTN_NADES_ redox peaks are shown in Figure 9a,b. Implementation of non-conductive DNA molecules into the surface layer led to the electron transfer hindrance and, therefore, peak currents decay.

Salmon sperm DNA-influenced peak currents are stronger than high-molecular calf thymus DNA. This could be due to better flexibility of low-molecular DNA from salmon sperm and the formation of more compact DNA layers as a result of biopolymer implementation. Incubation of the PTN-modified electrode in the solution of the salmon sperm DNA was chosen as optimal for the following experiments. A formal increase in the current over 100% (B column in Figure 9b) resulted from a higher deviation of the peaks on voltammograms obtained after the DNA drying. Taking into account the standard deviation presented in error bars, the peak current was not changed against the monomer dye signal referred to as 100%.

### 3.6. Impedimetric Epirubicin Determination

EIS offers broad opportunities both for the monitoring of the assembling of surface layers and for the measurements of recognition events with electrochemical biosensors [44]. The high sensitivity of the signal, mostly of the charge transfer resistance to the permeability of the surface layer and its charge, makes EIS attractive for the detection of various recognition events, e.g., hybridization of complementary DNA strands, DNA damage [45] and DNA intercalation [46]. Previously, we showed that, for the DNA–drug interaction, EIS parameters were more sensitive to the analytes than voltammetric techniques [47,48,49,50].

The possibility to detect epirubicin with the DNA sensor based on a polythionine layer with adsorbed DNA molecules follows from the mechanism of the DNA intercalation resulted in significant changes in the DNA shape, size, and surface charge distribution. All of these processes can be detected by the shift of the redox equilibrium of the polythionine support and appropriate changes in the electrochemical characteristics of the DNA sensor. Other species present in the samples tested can only affect this reaction non-specifically by adsorption or partial shielding of the biopolymer charge. Being reversible, such reactions can be rather easily eliminated by sample dilution. The efficiency of intercalations is caused by a planar polyaromatic core of the drug molecule, which was specifically designed for binding DNA as a biological target.

The interaction of the epirubicin with DNA leads to the formation of the drug–DNA complex, in which the planar ring system of epirubicin intercalates the DNA double-strand helix with further strand cleavage, the inhibition of DNA and RNA synthesis, and the inhibition of topoisomerase-II enzyme activity [51]. DNA intercalation changes the charge distribution and thickness of the modifying layer, affecting the accessibility of the electrode for the redox probe in the EIS measurements. The optimal incubation time for DNA–epirubicin interaction was tested in a series of experiments (Appendix A) and established as 10 min. Six sensors based on PTN_NADES_/DNA were used to evaluate the signal repeatability.

An equimolar mixture of 10 mM [Fe(CN)_6_]^3−^ and [Fe(CN)_6_]^4−^ ions was used as a redox probe. A half-sum of the cathodic and anodic potentials of the [Fe(CN)_6_]^3−/4−^ mixture equal to 0.11 V was chosen as the equilibrium potential, and the charge transfer resistance was applied for quantification of the DNA–epirubicin interactions. Typical Nyquist diagrams obtained during the DNA-sensor assembly and incubation of the DNA sensor in epirubicin solution are presented in Figure 10. 

Semicircles correspond to the limiting step of electron transfer on two interfaces, i.e., between the electrolyte and outer surface of the modifying layer and the second one between the inner part of the modifying layer and the electrode. Among them, the outer interface showed remarkable changes in the parameters with increasing concentration of the drug. This coincides well with the mechanism proposed because the outer interface contacts with a higher concentration of the drug than the inner interface.

The linear range of the epirubicin concentration based on the charge transfer resistance measurements was between 1.0 and 100 µM with the LOD 300 nM (S/N = 3, Figure 11a). An appropriate equation is presented in Equation (2).
*R*_1_, kΩ = (5.03 ± 0.11) + (0.61 ± 0.02) × log(*c*, M), R^2^ = 0.995, *n* = 6(2)

Although the cardiotoxicity of epirubicin is much lower than that of doxorubicin, about 10% of patients suffer from side effects and up to 5% within one year after completion of the chemotherapy [52]. In the case of intravenous administration, urinary excretion results in the removal of about 20% of the drug, mostly in the unmodified form. In total, 50% of epirubicin is eliminated within 4 days [53]. For the dose of 50 mg/m^2^ (1.74 m^2^ corresponds to 70 kg of body weight), the concentration of unchanged drug in plasma varied from 500 to 40 mg/L (0.92 mM–7 µM) within 1–40 h after intravenous administration [54] and to about 100 ng/mL (180 nM) after bolus administration [55]. Thus, the concentrations determined by the DNA sensor developed allow reliable monitoring of the drug dosage during the course and make it possible to control the dosage and diminish possible side effects for patients. 

The characteristics of epirubicin determination are comparable with other electrochemical sensors (Table 3). 

In most of them, direct or mediated oxidation of the drug was quantified after its accumulation on the modifier. Appropriate peaks on voltammograms were mostly observed at rather high anodic potentials, and many substances contained in the samples and medications can interfere with them. Modifiers mostly involve nanomaterials that were synthesized and applied to the electrodes in the time- and labor-consuming multi-stage procedures. The use of the DNA nucleotide oxidation signals [27] is the only alternative approach to drug oxidation. However, in this work, most attention was paid to the DNA damage caused by the anthracycline drugs, with a rather short description of the drug’s determination. Contrary to the voltammetric sensors, the use of the EIS protocol proposed in this work allowed measuring direct interactions of the drug with DNA in much milder conditions. The DNA-containing surface layer can be considered a simple but reliable model of target interactions taking place in chemotherapy. Application of NADES offers a simple and reproducible single-step way for the SPE modification using only one drop of the reaction media. The SPE construction also has minimal requirements to the conditions of the real assay with only one drop (100 µL) of the sample. This might be important for routine screening of the drug residues performed in POCT format.

The precision of signal measurement was evaluated using six individual DNA sensors prepared from the same set of reagents. Both the EIS parameters of the DNA sensors and their response toward 1 µM epirubicin were assessed. For the freshly prepared set of DNA sensors, sensor-to-sensor repeatability was equal to 4.5% (the standard deviation of the signal is presented here and below). We did not find any drift of the signal when the period between the DNA sensor assembling and epirubicin incubation varied from several hours to two days. Each DNA sensor was used for a single measurement. After three weeks of storage in dry conditions, the deviation of the signals increased to 6.5%. It is interesting to note that the stability of the signal of the PTN_NADES_/DNA sensor was even better than that obtained from HEPES (PTN_aq_/DNA sensor). For the latter one, the standard deviation of the signals was 5.5 and 7.7% prior to and after two weeks of storage.

### 3.7. Real Sample Impedimetric Analysis

To evaluate the applicability of the developed sensor to real sample analysis, several artificial and real biological samples were tested. Ringer-Locke’s solution, bovine serum albumin solution, artificial urine, human urine, and human saliva (Table 4) were intentionally spiked with epirubicin until the final concentration was equal to 10 μM. Urine contains urea, creatinine, amino acids, uric acid, ketone bodies, creatine, glucose, hippuric acid, and inorganic cations and anions [56]. Human saliva is a complex mixture of various electrolytes and immunoglobulins, proteins, enzymes, mucin, urea, and ammonia [57,58]. Dilution with a working buffer can suppress the matrix effects. We chose minimal dilution (1:1) with the HEPES buffer and found satisfactory correspondence of the peaks recorded in the spiked samples and standard solution (1.94 kΩ). This made it possible to conclude that inorganic electrolytes commonly contained in plasma, glucose, urea, and lactate did not affect the EIS characteristics of the PTN_NADES_/DNA sensor.

No significant influence of the BSA solution and artificial and real urine samples were found either. The recovery of epirubicin detection assessed against its standard solution was slightly improved after 1:1 dilution with HEPES due to the prevention of the obstructive influence of plasma electrolytes and microheterogeneity of the sample.

We also performed HPLC determination of epirubicin in the spiked samples of human saliva as that with the most complex matrix among others tested. For 10 μM epirubicin, the concentration found was equal to 9.8 ± 0.2 μM (recovery 98%). No drug was found in blank experiments with human saliva from healthy volunteers. 

It should be noted that other anthracycline drugs also interact with DNA and will affect the signals of the DNA sensors based on electropolymerized redox active matrices. It was shown earlier for doxorubicin, daunorubicin, and idarubicin in polyaniline [59] and idarubicin in the polyphenothiazine dye [60] layer. Such similarity of response follows from a general mechanism of DNA–drug interaction. Nevertheless, the drug administration assumes mostly only one anthracycline medication applied with auxiliary substances of other nature. For this reason, there is no possibility of finding complications in the DNA sensor signal interpretation for cancer treatment cases.

### 3.8. Conclusions

In this work, the PTN electropolymerized from NADES consisting of citric acid monohydrate, glucose, and water in a molar ratio of 1:1:6 was, for the first time, applied as a signal-forming agent and matrix for DNA adsorption. The resulting layer attached to the carbon SPE showed sensitivity toward a model intercalator drug, epirubicin, belonging to the anthracycline family of antitumor medications. Both the formation of the biorecognition layer and incubation in the epirubicin-containing sample were performed at physiological pH and in the mild conditions that did not assume necessity in organic solvents both for monomeric thionine dissolution and an analyte extraction. The protocols of DNA sensor assembling and signal detection are performed quickly and with minimal treatment of the electrode/sample. The application of planar SPEs allows drop casting of 100 µL of the mixtures to establish hydrolytic contact between the electrodes. The comparison of the electrochemical properties of polymeric layers obtained in the NADES and in common conditions (aqueous HEPES) made it possible to conclude the similarity of the mechanisms of their synthesis and deposition. Meanwhile, the PTN_NADES_/DNA sensor showed more irregular morphology of the surface caused probably by a higher viscosity of the media and slower access of the monomer to the electrode interface during the potential scanning. The assembling of the DNA sensor from NADES minimized the consumption of the reactants and the quantities of the wastes. This makes the approach proposed very attractive for the manufacture of point-of-care devices, especially intended for use outside well-equipped laboratories and the developing countries with limited healthcare services. 

The impedimetric signal diminishes the potential of electrode polarization and prevents interference with electroactive substances present in the medication and biological fluids. The sensitivity of epirubicin determination is comparable with the performance of electrochemical sensors based on mediated oxidation of the drug requiring higher potentials and rather sophisticated synthesis of nanomaterials used for electrode modification. Although the concentrations determined with the DNA sensor based on the approach presented are slightly higher than those reported for the analogs, they cover the drug levels typical for the cancer patient’s administration. Thereby, the derived biosensors can find application in medicine, pharmacokinetic assays, and the control of medical substances in drugs.

## Figures and Tables

**Figure 1 sensors-23-08242-f001:**
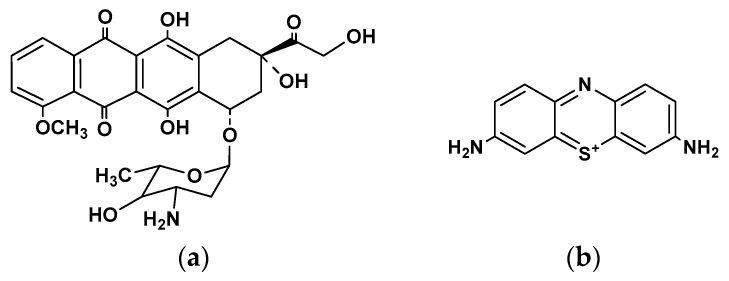
Chemical structures of (**a**) epirubicin and (**b**) thionine monomer (oxidized form).

**Figure 2 sensors-23-08242-f002:**
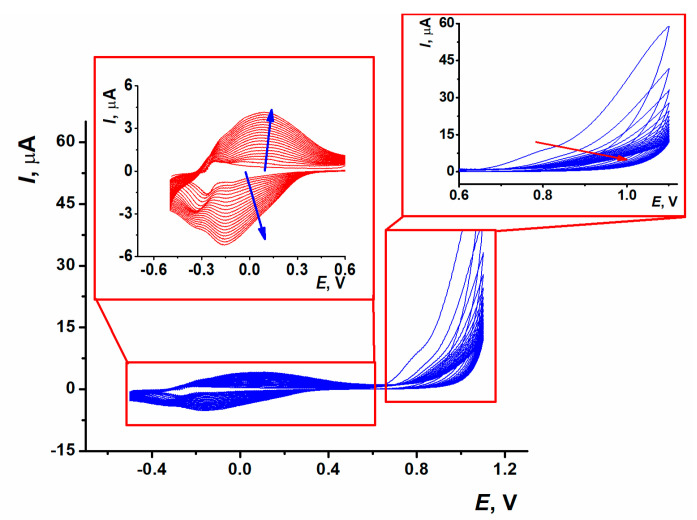
Multiple cyclic voltammograms recorded on the carbon SPE in 0.1 M HEPES containing 0.03 M NaCl and 0.1 mM thionine, pH 7.0, scan rate 100 mV/s, 20 cycles. Insets: enlarged parts corresponding to the peaks of polymeric and monomeric thionine (**left**) and cation radical formation (**right**) on the voltammogram. Arrows indicate direction of the changes with increasing number of potential cycles.

**Figure 3 sensors-23-08242-f003:**
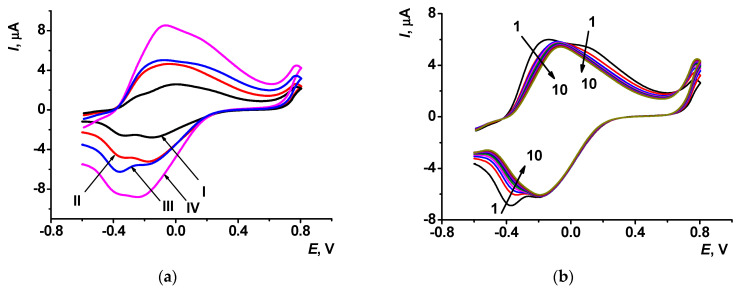
Cyclic voltammograms recorded on the SPE in 0.1 M HEPES containing 0.03 M NaCl, pH 7.0; scan rate 100 mV/s: (**a**) electropolymerization from aqueous solution of the monomer, 7th (I), 15th (II), 20th (III), and 30th (IV) cycles of the potential; (**b**) cyclic voltammograms of the PTN_aq_ layer (10 consecutive cycles) after its stabilization in open-circuit mode. Arrows indicate changes from the first to tenth cycle.

**Figure 4 sensors-23-08242-f004:**
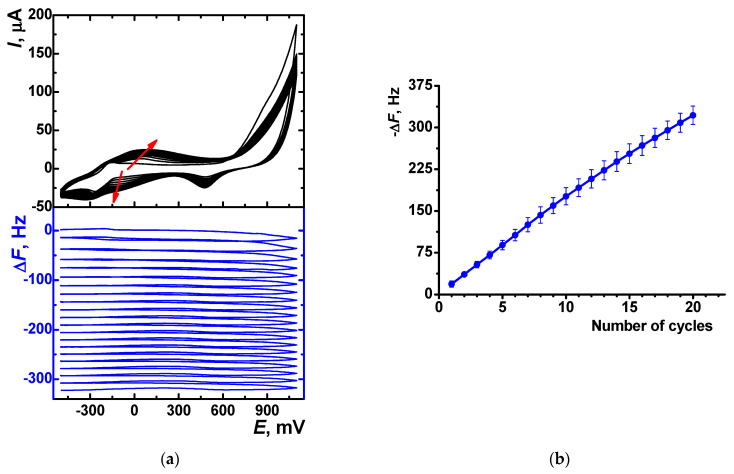
(**a**) Cyclic voltammograms (black) and sensograms (blue) recorded on the Au QCM chip in 0.1 M HEPES containing 0.1 mM thionine and 0.03 M NaNO_3_, pH = 7.0. Arrows indicate the changes with increased number of cycles; (**b**) EQCM frequency shift with the number of cycles of potential during the thionine electropolymerization, average ± S.D. from six measurements.

**Figure 5 sensors-23-08242-f005:**
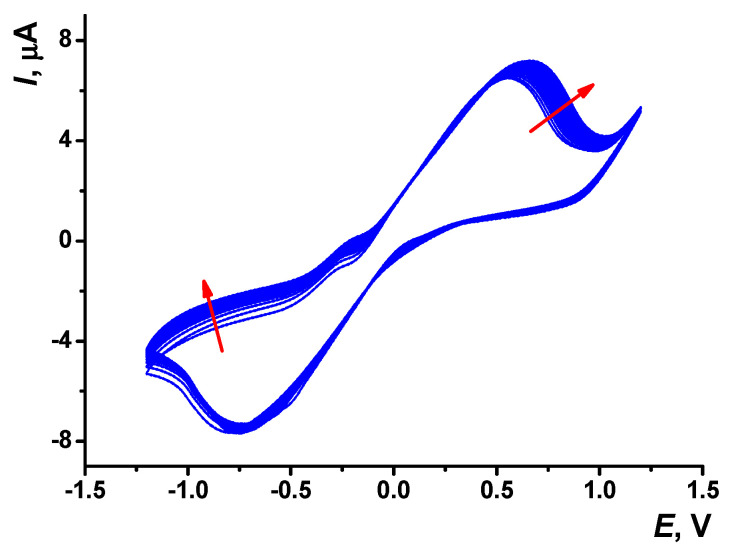
Multiple cyclic voltammograms recorded on the carbon SPE in NADES containing 0.1 M thionine, scan rate 100 mV/s, 20 cycles. Arrows indicate the changes with increased number of cycles.

**Figure 6 sensors-23-08242-f006:**
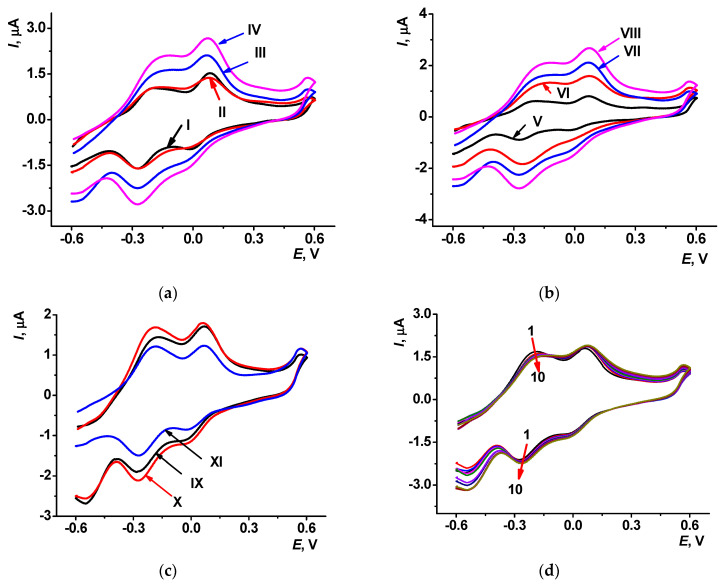
Cyclic voltammograms recorded at the scan rate of 100 mV/s on the carbon SPE modified with PTN_NADES_ in 0.1 M HEPES containing 0.03 M NaCl, pH 7.0; (**a**) after 7th (I), 15th (II), 20th (III), and 30th (IV) cycles of the potential, 0.067 M thionine; (**b**) after electropolymerization of 0.017 (V), 0.034 (VI), 0.067 (VII) and 0.1 M (VIII) thionine, 20 cycles; (**c**) after 20 cycles of 0.1 M thionine electropolymerization from NADES, scan rate 50 (IX), 100 (X), and 300 mV/s (XI); (**d**) PTN_NADES_ stability evaluation in 10 consecutive cycles performed in 0.1 M HEPES containing 0.03 M NaCl, pH 7.0, scan rate 100 mV/s. Arrows indicate changes with increased number of cycles.

**Figure 7 sensors-23-08242-f007:**
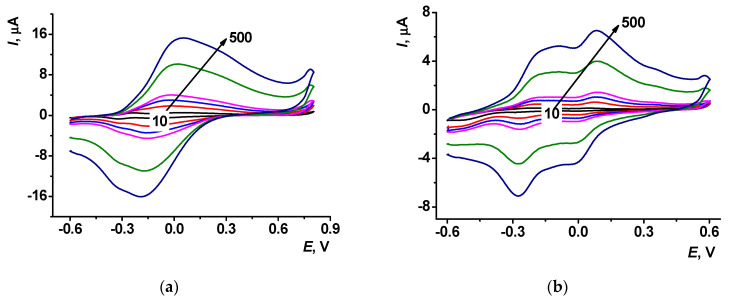
(**a**) Cyclic voltammograms recorded on (**a**) PTN_aq_ and (**b**) PTN_NADES_ modified SPE in 0.1 M HEPES containing 0.03 M NaCl, pH 7.0, at the scan rates of 10, 40, 70, 100, 300, and 500 mV/s.

**Figure 8 sensors-23-08242-f008:**
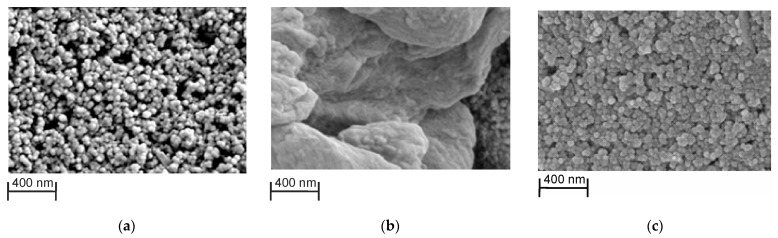
SEM images of the (**a**) bare carbon SCE; (**b**) covered with PTN_NADES_; (**c**) covered with PTN_aq._

**Figure 9 sensors-23-08242-f009:**
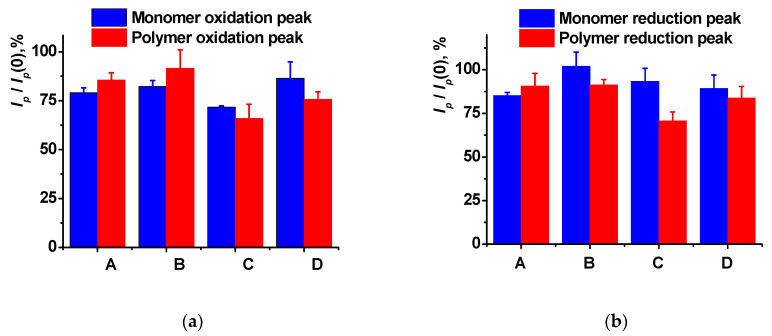
Relative changes in the peak currents (prior to contact with DNA *I_p_*(0), after the DNA inclusion *I_p_*) of the PTN_NADES_ resulted from 20 min. Incubation (A, C) and drying the DNA drop (B, D) of the DNA from salmon sperm (A, B) and calf thymus (C, D); (**a**) oxidation peaks; (**b**) reduction peaks. Cyclic voltammetry, 0.1 M HEPES containing 0.03 M NaCl, pH 7.0, scan rate 100 mV/s. Average ± S.D. for six individual sensors.

**Figure 10 sensors-23-08242-f010:**
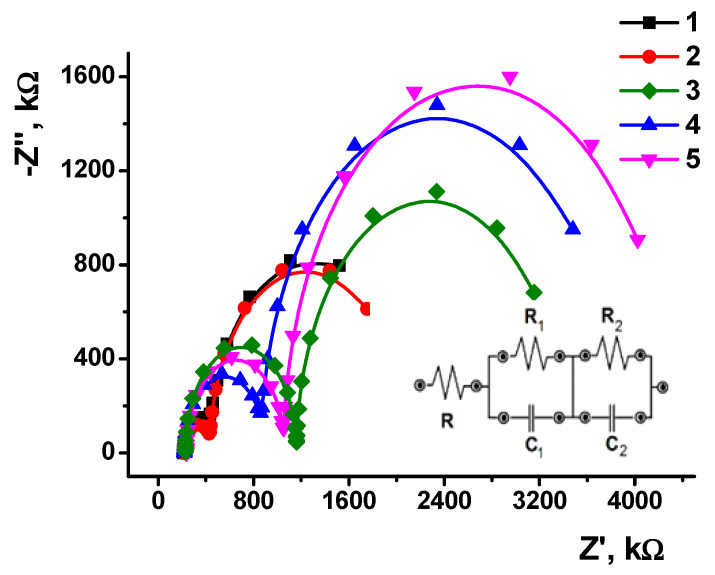
The Nyquist diagrams recorded for PTN_NADES_ (1), PTN_NADES_/DNA (2), PTN_NADES_/DNA/Epirubicin (10 µM) (3), PTN_NADES_/DNA/Epirubicin (50 µM) (4), and PTN_NADES_/DNA/Epirubicin (100 µM) (5). Inset: Equivalent circuit applied for EIS data fitting. *R* is the solvent resistance, and *R*_1_ (*R*_2_) and *C*_1_ (*C*_2_) are the charge transfer resistance and constant phase elements, respectively. Indices ‘1′ and ‘2′ correspond to the electrolyte–surface layer and surface layer–electrode interfaces^.^

**Figure 11 sensors-23-08242-f011:**
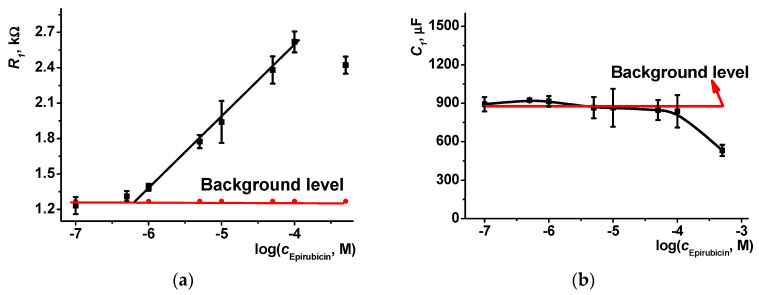
The dependence of the EIS parameters on the epirubicin concentration; (**a**) charge transfer resistance *R*_1_, (**b**) constant phase element *C*_1_. PTN_NADES_/DNA sensor, 0.1 M HEPES containing 0.03 M NaCl, pH 7.0, 0.01 M [Fe(CN)_6_]^3−/4−^. Average ± S.D. for six individual sensors.

**Table 1 sensors-23-08242-t001:** Slopes of the linear part of the *E_m_*-pH dependency for the PTN_NADES_ and PTN_aq_ films.

Polymer	Slope d*E_m_*/dpH, V/pH	Linear pH Range
PTN_aq_	−0.079 ± 0.008	2–6
PTN_NADES_ (monomer)	−0.055 ± 0.005	3–8
PTN_NADES_ (polymer)	−0.066 ± 0.003	2–9

**Table 2 sensors-23-08242-t002:** Electrochemical parameters of the PTN_NADES_ and PTN_aq_ coatings.

Polymer/Peak on Voltammogram	dlog*I_p_*/dlogν (Linear Range, V/s)	d*E_p_*/dlogν (Linear Range, V/s)	Electron Transfer Coefficient
PTN_aq_/oxidation peak	0.81 ± 0.01 (0.01–0.5)	0.071 ± 0.005 (0.01–0.05)	(1 − α) = 0.83
PTN_aq_/reduction peak	0.84 ± 0.02 (0.01–0.5)	−0.038 ± 0.007 (0.1–0.5)	α = 1.55
PTN_NADES_/monomer oxidation peak	0.79 ± 0.01 (0.02–0.5)	0.024 ± 0.005 (0.01–0.4)	(1 − α) = 2.47
PTN_NADES_/polymer oxidation peak	0.78 ± 0.01 (0.02–0.5)	0.033 ± 0.001 (0.01–0.08)	(1 − α) = 1.81
PTN_NADES_/monomer reduction peak	1.12 ± 0.02 (0.02–0.5)	−0.178 ± 0.001 (0.09–0.5)	α = 3.3
PTN_NADES_/polymer reduction peak	0.91 ± 0.01 (0.01–0.5)	−0.189 ± 0.001 (0.08–0.5)	α = 3.12

**Table 3 sensors-23-08242-t003:** Epirubicin determination with electrochemical sensors ^1.^

Electrode/Modifying Layer	Signal Measurement Conditions	Linear Range, LOD	Ref.
GCE covered with Ag/MWCNTs composite	Oxidation peak current, SWV	0.003–0.25 μM, 1.0 nM	[23]
SPE covered with MoS_2_/PEDOT:PSS dispersion	Oxidation peak current, DPV	0.06–9.30 μM, 44.3 nM	[24]
Pencil graphite working electrode covered with WS_2_/PEDOT:PSS dispersion	Oxidation peak current, DPV	0.172–345 μM, 35 nM	[26]
SPE modified with Au@NiFe_2_O_4_ nanoparticles	Oxidation peak current, DPV	0.01–3.6 µM, 5.3 nM	[27]
SPE modified with dsDNA/Pt NPs/ Ag NPs/SPE	Guanine and adenine oxidation peak currents, DPV	0.05–1 ppm (0.09–1.8 μM)	[28]
SPE modified with PTN obtained from NADES	Charge transfer resistance, EIS	1.0–100 μM, 300 nM	This work

^1^ Acronyms used in Table 3: GCE—glassy carbon electrode, MWCNTs—multi-walled carbon nanotubes, PEDOT—poly(ethylene dioxythiophene), PSS—poly(styrene sulfonate), SPE—screen-printed electrode, dsDNA—double-stranded deoxyribonucleic acid, NPs—nanoparticles, PTN—polythionine, EIS—electrochemical impedance spectroscopy, SWV—square wave voltammetry, DPV—differential pulse voltammetry.

**Table 4 sensors-23-08242-t004:** Charge transfer resistance *R*_1_, kΩ (1.94 kΩ in standard solution of the drug), and recovery measured in spiked samples of artificial serum, human plasma, and urine, as well as of real human urine and saliva. Charge transfer resistance *R*_1_ after incubation in 10 μM epirubicin changes. Average ± S.D. for six individual sensors.

Sample	*R*_1_, kΩ (10 µM Epirubicin)	Recovery, %
Ringer-Locke’s solution	1.76 ± 0.08	90.5 ± 4.3
Ringer-Locke’s solution (1:1 dilution with HEPES)	1.98 ± 0.13	101.8 ± 6.6
BSA 41.4 mg/mL	1.98 ± 0.14	102.0 ± 7.2
Artificial urine	1.97 ± 0.11	101.7 ± 5.9
Human urine 1	2.09 ± 0.10	107.8 ± 4.9
Human urine 2	2.11 ± 0.07	108.9 ± 3.6
Human saliva 1	1.48 ± 0.08	76.5 ± 3.9
Human saliva 1 (1:1 dilution with HEPES)	2.04 ± 0.19	105.3 ± 9.8
Human saliva 2 (1:1 dilution with HEPES)	2.03 ± 0.13	104.7 ± 6.5

## Data Availability

The data presented in this study are available in Appendix A.

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
