# Peer review of "Impedimetric DNA Sensors for Epirubicin Detection Based on Polythionine Films Electropolymerized from Deep Eutectic Solvent"

_sensors, 2023, doi:10.3390/s23198242_

Round 1

Reviewer 1 Report

The authors proposed an Impedimetric DNA-Sensor for Epirubicin Detection Based on Polythionine Films Electropolymerized from Deep Eutectic Solvent. The subject of this work is not very prominent and it is not clear whether the authors are emphasizing the preparation of materials or the preparation of biosensors for epirubicin detection. It is seen that this work has been effective, but there are still some problems that need to be improved

1. The theme of the manuscript is not very prominent, the theme should be clear.

2. Please clearly underline the novelty of this work in context to the previous findings of other articles.

3. How can be explained the selectivity of epirubicin to the proposed method?

4. The author should explain why the impedance method is chosen over other sensitive techniques such as DPV or SWV.

5. It seems that the calibration curve needs to be rechecked because it is not linear and does not match with the R2=0.995.

6. The detection results for real samples should be validated by the other analytical techniques.

7. The quality of presentation, in terms of usage of the English language, needs to be improved.

Reviewer 2 Report

The presented research article describes the design, creation, and characterization of an impedimetric DNA sensor for quantitative detection of anti-cancer drug epirubicin in biological samples. The task is very timely and interesting since up to date oncologists still lack convenient point-of-care sensors for monitoring the course of chemotherapy. The article has strong background and very thorough electrochemical methodology. It is logically structured and well-written. I would recommend it to publication after adding some minor changes.

The authors did not mention the influence of other DNA intercalating drugs or natural DNA intercalators (e.g., riboflavin) on the results of detection. However, this point seems to be very important when we transfer the test system to the analysis of real blood, urine or saliva samples. I suggest to add at least the discussion of this topic either to the Subsection 3.7 or to the Conclusion.

Author Response

The presented research article describes the design, creation, and characterization of an impedimetric DNA sensor for quantitative detection of anti-cancer drug epirubicin in biological samples. The task is very timely and interesting since up to date oncologists still lack convenient point-of-care sensors for monitoring the course of chemotherapy. The article has strong background and very thorough electrochemical methodology. It is logically structured and well-written. I would recommend it to publication after adding some minor changes.

The authors did not mention the influence of other DNA intercalating drugs or natural DNA intercalators (e.g., riboflavin) on the results of detection. However, this point seems to be very important when we transfer the test system to the analysis of real blood, urine or saliva samples. I suggest to add at least the discussion of this topic either to the Subsection 3.7 or to the Conclusion.

We agree with esteemed Reviewer that interaction of DNA with other intercalating drugs is important and could affect the sensor signal. Earlier such an influence was found for doxorubicin and idarubicin as representatives of the anthracycline family. However, protocols of drug administration assume the use of a single medication so that possibility to found two anthracyclines in the blood serum of the patients is insignificant. We suggest the following text to be added to the Section 3.7:

It should be noted that other anthracycline drugs also interact with DNA and will affect the signals of the DNA sensors based on electropolymerized redox active matrices. It was shown earlier for doxorubicin, daunorubicin and idarubicin in polyaniline [59] and idarubicin in polyphenothiazine dyes [60] layer in polyaniline and idarubicin in polyphenothiazine dyes layer. Such similarity of response follows from a general mechanism of DNA-drug interaction. Nevertheless, the drug administration assumes mostly only one anthracycline medication applying with auxiliary substances of other nature. For this reason, there is no possibilities to find complications in the DNA sensor signal interpreting for cancer treatment cases.

  1. Shamagsumova, R.; Porfireva, A.; Stepanova, V.; Osin, Y.; Evtugyn, G.; Hianik, T. Polyaniline–DNA based sensor for the detection of anthracycline drugs. Sens. Actuators B, 2015, 220, 573–582. https://doi.org/10.1016/j.snb.2015.05.076.
  2. Evtugyn, G.; Porfireva, A.; Stepanova, V. Budnikov, H. Electrochemical biosensors based on native DNA and nanosized mediator for the detection of anthracycline preparations, Electroanalysis, 2015, 27, 629-637. https://doi.org/10.1002/elan.201400564

Reviewer 3 Report

The research activity described in the manuscript appears well organized and with an appropriate level of originality for its publication in the journal. The manuscript is also well organized and clearly explains the experimental activity carried out and the results obtained. There remain only a few minor issues to resolve in order to have a version of the manuscript ready for publication.

Below is the list of aspects to be resolved:

1 - Page 2 row 46: a reference on NADES cytotoxicity would be desirable;

2 - Page 3 rows 95/96: replace the term "sensos" with the term "sensors";

3 - Page 5 Figure 2: add arrows to indicate changes with increasing number of cycles and a second inset with a zoom of the high voltage side of the voltammogram;

4 - Page 11 Figure 9.b: a comment related to the relative changes value (greater than 100%) reported for the monomer reduction peak obtained for the (B) configuration is required;

5 - Page 12 row 397: replace "(6)" with "(5)";

6 - Page 13 row 451 and Page 14 rows 455 and 457: the percentage changes reported are referred to the coefficient variation? More details are required;

7 - Page 14 row 469: replace "," with "." and replace "M" with "uM";

8 - Page 14 Table 4: further comments are needed on the results reported in Table 4, in particular regarding the recovery,% values in the case of the samples Ringer-Locke's solution and Human Saliva 1.

Round 2

Reviewer 1 Report

Accept in present form